# High risk oral contraceptive hormones do not directly enhance endothelial cell procoagulant activity *in vitro*

Emma G. Bouck[1], Marios Arvanitis[2], William O. Osburn[2], Yaqiu Sang[1], Paula Reventun[2], Homa K. Ahmadzia[3], Nicholas L. Smith[4,5,6], Charles J. Lowenstein[2], Alisa S. Wolberg[1]*

1 Department of Pathology and Laboratory Medicine and UNC Blood Research Center, University of North Carolina at Chapel Hill, Chapel Hill, NC, United States of America, 2 Department of Medicine, The Johns Hopkins University School of Medicine, Baltimore, MD, United States of America, 3 Division of Maternal-Fetal Medicine, Department of Obstetrics and Gynecology, George Washington University, Washington, DC, United States of America, 4 Department of Epidemiology, University of Washington, Seattle, WA, United States of America, 5 Kaiser Permanente Washington Health Research Institute, Kaiser Permanente Washington, Seattle, WA, United States of America, 6 Seattle Epidemiologic Research and Information Center, Department of Veterans Affairs Office of Research and Development, Seattle, WA, United States of America

* alisa_wolberg@med.unc.edu

**Data Availability Statement:** All relevant data are within the paper and its Supporting Information files.

## Abstract

### Background

Oral contraceptive (OC) use increases venous thromboembolism risk 2-5-fold. Procoagulant changes can be detected in plasma from OC users even without thrombosis, but cellular mechanisms that provoke thrombosis have not been identified. Endothelial cell (EC) dysfunction is thought to initiate venous thromboembolism. It is unknown whether OC hormones provoke aberrant procoagulant activity in ECs.

### Objective

Characterize the effect of high-risk OC hormones (ethinyl estradiol [EE] and drospirenone) on EC procoagulant activity and the potential interplay with nuclear estrogen receptors ERα and ERβ and inflammatory processes.

### Methods

Human umbilical vein and dermal microvascular ECs (HUVEC and HDMVEC, respectively) were treated with EE and/or drospirenone. Genes encoding the estrogen receptors ERα and ERβ (*ESR1* and *ESR2*, respectively) were overexpressed in HUVEC and HDMVEC via lentiviral vectors. EC gene expression was assessed by RT-qPCR. The ability of ECs to support thrombin generation and fibrin formation was measured by calibrated automated thrombography and spectrophotometry, respectively.

**Funding:** This study was funded by grants from the National Institutes of Health (https://www.nih.gov; R61HL141791, R33HL141791 to ASW/NLS/CJL; R01HL126974 ASW; T32GM122741 to ASW/EGB; T32HL069768 to UNC/EGB). The funders had no role in study design, data collection and analysis, decision to publish, or preparation of the manuscript.

**Competing interests:** The authors have declared that no competing interests exist.

## Results

Neither EE nor drospirenone, alone or together, changed expression of genes encoding anti- or procoagulant proteins (*TFPI*, *THBD*, *F3*), integrins (*ITGAV*, *ITGB3*), or fibrinolytic mediators (*SERPINE1*, *PLAT*). EE and/or drospirenone did not increase EC-supported thrombin generation or fibrin formation, either. Our analyses indicated a subset of individuals express *ESR1* and *ESR2* transcripts in human aortic ECs. However, overexpression of *ESR1* and/or *ESR2* in HUVEC and HDMVEC did not facilitate the ability of OC-treated ECs to support procoagulant activity, even in the presence of a pro-inflammatory stimulus.

## Conclusions

The OC hormones EE and drospirenone do not directly enhance thrombin generation potential of primary ECs *in vitro*.

## Introduction

Oral contraceptives (OCs) are used by ~100 million women worldwide to prevent unplanned pregnancies and to treat hormone-mediated disorders (e.g., polycystic ovarian syndrome) [1]. Access to OCs reduces maternal mortality by 44%, and optimizing access to contraception could further reduce mortality rates by 29% across 172 countries [2]. Access to contraception including OCs is correlated with increased education rates for women and decreased likelihood of living in poverty across generations [3]. However, OC use is associated with 2-5-fold increased venous thromboembolism (VTE) risk [4–6], such that 1 in 3,000 OC users develop VTE annually (over 30,000 individuals per year). Uncovering mechanisms that mediate OC-induced VTE would more safely avert unwanted pregnancies, hormonal imbalances, and preventable deaths.

Since their introduction in the 1960s, OC formulations have undergone several iterations to reduce VTE risk and other side effects, including weight gain and mood changes. Modern OC formulations contain both ethinyl estradiol (EE) and a progestin, because of the generally opposing biological effects of these two hormones. Progestins prevent pregnancy but can cause side effects. EE counteracts these side effects; however, EE dose is positively correlated with VTE risk [5, 7, 8]. VTE risk, in turn, is modified by the type of progestin used [9, 10]. Although controversial, it is generally appreciated that third and fourth generation progestins carry higher VTE risk than earlier OC formulations [6, 11]. Drospirenone, a fourth generation progestin, increases VTE risk 6.3-fold compared with no OC use, and 2-3-fold compared with lower risk progestins, such as levonorgestrel [5, 12, 13]. However, the mechanisms by which EE and progestins contribute to VTE are poorly understood [4–6].

VTE pathogenesis involves a combination of defects in blood flow, blood procoagulant activity, and endothelial function (Virchow's Triad) [14]. Previous studies have identified changes in plasma composition and procoagulant activity induced by OC use. For example, plasma from OC users contains increased procoagulant proteins, including prothrombin, fibrinogen, and factor (F)VII, FVIII, and FX [15–22], and decreased anticoagulant proteins including antithrombin and protein S [17, 20, 21, 23, 24] Plasma from OC users shows enhanced resistance to the activated protein C anticoagulant pathway [25]. Plasma from OC users also has altered fibrinolytic pathway proteins, including increased plasminogen, tissue plasminogen activator, and plasmin-antiplasmin complexes, and decreased plasminogen

activator inhibitor type I (PAI-1). [24, 26]. These shifts in plasma composition are detectable within the first 3 months of OC use, indicating changes induced by OCs happen relatively quickly [5]. However, these changes are also routinely observed in OC users without VTE, suggesting abnormalities in plasma composition and function are insufficient to cause OC-related VTE [23]. Mouse models have been used to investigate systemic effects of OCs leading to thrombosis but have unexpectedly yielded antithrombotic responses [27–33]. These findings necessitate the use of human-derived *in vitro* models to probe OC-induced changes in other pathogenic components of VTE.

Endothelial cells (ECs) generally have an anticoagulant phenotype through expression of proteins such as tissue factor pathway inhibitor (TFPI), which inhibits the FVIIa/tissue factor complex and blocks coagulation initiation, and thrombomodulin, which supports thrombin-mediated activation of protein C and blocks coagulation propagation [34–37]. ECs also synthesize tissue plasminogen activator and PAI-1, which mediate fibrinolytic activity to limit fibrin deposition [38, 39]. However, when provoked by inflammatory stimuli, ECs become activated and may express tissue factor which initiates coagulation, integrins including $\alpha_v\beta_3$ which may influence fibrin network organization, and/or alter expression of anticoagulant or fibrinolytic proteins [37, 40–43]. The ability of EE and progestins to directly stimulate EC procoagulant or fibrinolytic activity has not been comprehensively investigated.

To determine whether OC hormones provoke EC prothrombotic activity, we designed an *in vitro* model of the plasma-endothelial interface. We treated ECs with EE and drospirenone (alone or together) and measured the effects of hormone treatments on transcriptional and procoagulant responses in native cells and in cells expressing genes encoding the canonical nuclear estrogen receptors (ERα and ERβ) and/or a proinflammatory stimulus. Our findings suggest EE and drospirenone do not directly induce prothrombotic activity in primary human ECs.

## Methods

### Plasma preparation

All methods involving human subjects were approved by the UNC Institutional Review Board (01–1274). Written consent was obtained from all subjects. To prepare normal, pooled, human plasma (NPP), whole blood from 20–30 donors was collected into corn trypsin inhibitor (18.6 ug/mL final, Haematologic Technologies, Inc) and sodium citrate (0.32% final). Platelet-poor plasma was prepared by sequential centrifugation: 20 minutes at 150x*g* to prepare platelet-rich plasma and 20 minutes at 20,000x*g* to remove platelets. A separate aliquot of platelet-poor plasma without corn trypsin inhibitor was prepared to measure each donor's activated partial thromboplastin time before inclusion in the pool. Fresh corn trypsin inhibitor-treated, citrated platelet-rich and platelet-poor plasma from individual donors was also prepared and used within 2 hours of blood collection.

### Cell culture and treatments

Human umbilical vein ECs (HUVEC) were purchased from PromoCell (Cat. C-12200, Lot 428Z011.3, newborn Caucasian female). Human dermal microvascular EC (HDMVEC) lots were purchased from Lifeline (Cat. FC-0039, Lot 04014, 42-year-old African American female) or Lonza (Cat. C-12212, Lot 19TL136480, 47-year-old Caucasian female). HUVEC and HDMVEC were cultured in phenol-red free VascuLife® VEGF-Mv Endothelial Complete Kit (Lifeline Cat. LL-0005) supplemented with 5% fetal bovine serum. Plates were coated with 50 μg/mL collagen (Sigma-Aldrich, Cat. C3867) in 20 mM acetic acid overnight at 37˚C. Plates were rinsed 3X with Dulbecco's phosphate-buffered saline (PBS) before plating cells. When

ECs reached 90% confluency, they were dissociated with 0.05% trypsin. For experimental plates, cells were resuspended at 50,000 cells/mL and the appropriate volume was added to each well (6-well plate, 2 mL; 96-well plate, 200 μL). EC media with fetal bovine serum was switched to media containing 2.5% charcoal-stripped serum ([CSS], Gibco, Cat. 12676029) 24 hours before hormone treatment.

MCF-7 cells (ATCC, Cat. HTB-22) were cultured in Eagle's Minimum Essential Media without phenol-red (VWR, Cat. 10128–658) supplemented with 10% fetal bovine serum, 0.01 mg/mL insulin (Gibco, Cat. 12585014), L-glutamine, and penicillin-streptomycin. MCF-7 media was switched to media containing 5% CSS 24 hours before hormone treatment.

EE was dissolved in 100% ethanol to 100 mM and stored at -20˚C; drospirenone was dissolved in 100% ethanol to 10 mM. Hormone stocks were diluted to 100X of final concentrations in 70% ethanol, then diluted 1:100 in CSS media. Tumor necrosis factor-α (TNFα, Sigma-Aldrich, Cat. T6674) stocks (100 μg/mL) were diluted to 100X of final concentrations in water, and then diluted 1:100 in CSS media. For time course experiments, the hormone-containing media was replaced every 24 hours. After hormone or TNFα treatment, cells in 6-well plates were harvested to prepare RNA or protein lysates, and 96-well plates were used for thrombin generation and fibrin formation assays.

## Reverse transcription-quantitative polymerase chain reaction (RT-qPCR)

Cells were lysed in the wells and RNA was extracted from cells using the Qiagen RNeasy kit, per manufacturer instructions. cDNA was prepared using the Qiagen QuantiTect Reverse Transcription Kit and diluted in 2 parts RNAse-free water (20 μL cDNA + 40 μL water). RT-qPCR reactions contained 2 μL cDNA, 1X SYBR green reagent, and 2 μM forward and reverse primers. Primers were obtained from Integrated DNA Technology (Coralville, Iowa); the primers used to read each transcript are listed in S1 Table in S1 File. Transcript amplification was measured using the QuantStudio3 system (Applied Biosystems). Geometric means of $C_T$ values for three housekeeping genes (*18S*, *SDHA*, *RPL13A*) were used to correct raw experimental $C_T$ values. Separately, cDNA was harvested from HDMVEC using RT$^2$ First Strand Kit (Qiagen, Cat 330404) and gene expression was assessed using a Human Endothelial Cell RT$^2$ Profile PCR Array (Qiagen, Cat. PAHS-015Z), per manufacturer instruction using the CFX96 Real-Time PCR Detection System (BioRad) and five housekeeping genes (*ACTB*, *B2M*, *GAPDH*, *HPRT1*, *RPLP0*). All relative expression values were calculated based on the $C_T$ value from untreated cells of the same passage.

## Thrombin generation assays

Cell supernatant was removed and cells were briefly rinsed with HEPES-buffered saline (HBS, 20 mM HEPES pH 7.4, 150 mM NaCl). HBS or thrombin generation calibrator (20 μL) and plasma (80 μL) were then added to each well. For experiments with NPP, plasma was diluted 1:1 in HBS. For experiments that included exogenous tissue factor, Siemens Dade™ Innovin™ was diluted 1:600 (final 1:9,000) and 20 μL was added to wells in lieu of the HBS. Thrombin generation was measured for 2 hours using the ThermoScientific Thrombinoscope instrument and software (version 5.0); parameters were calculated as described [44, 45].

## Fibrin formation assays

Cell supernatant was removed and cells were briefly rinsed with HBS. NPP was diluted 1:1 in HBS and recalcified (10 mM final) immediately before adding to cells. Fibrin formation was measured via absorbance at 405 nm for 2 hours using a Spectra Max Plus 384 spectrophotometer.

### HAEC RNA-seq data analysis

RNA-seq reads from 53 unique individuals were obtained from the NCBI Genbank (GEO accession number: GSE139377) [46]. Raw reads were processed using the ENCODE DCC RNA-seq analysis pipeline [47]. Reads were mapped to the hg38 version of the human genome using the STAR aligner v2.4.2 and reads that overlap exonic regions were aggregated using RSEM v1.2.31 with Gencode v29 as reference to produce a transcripts per million (TPM) matrix. Boxplots of gene TPM were generated for coagulation and estrogen receptor genes.

### Dot blot

Whole cell lysates were prepared in RIPA buffer (RPI, Cat. R26200) with protease and phosphatase inhibitor (Cell Signaling Technologies, Cat. 5827S). Protein concentration was determined using the Pierce™ BCA protein assay (Thermo Scientific, Cat. 23227) and lysates were normalized. Lysate and protein samples (diluted in RIPA) were loaded onto a pre-wetted nitrocellulose membrane using the BioRad BioDot™ Apparatus and rinsed with TBST before dissembling. Membranes were incubated in Intercept® Blocking Buffer (LI-COR, Cat. 927–60001) for 1 hour, and incubated overnight in primary antibody at room temperature. ERα was detected with a rabbit polyclonal antibody (AbCam, Cat. ab75635) in combination with D8H8 rabbit monoclonal antibody (Cell Signaling Technologies, Cat. 8644) and ERβ was detected with 14C8 mouse monoclonal antibody (GeneTex, Cat. AB_370367), followed by IRDye® 800CW Goat-anti-Rabbit secondary antibody (LI-COR, Cat. 926–32211) and IRDye® 680RD Goat-anti-Mouse secondary antibody (LI-COR, Cat. 926–68070), respectively. The membranes were imaged using the Bio-Rad Chemi-Doc™ MP imaging system.

### Lentiviral expression of *ESR1* and *ESR2*

Bacterial plasmids encoding *ESR1* (Cat. VB900122-1582cgb), *ESR2* (Cat. VB900083-8463kfc), or mCherry (Cat. VB010000-9298rtf) were purchased from VectorBuilder. Plasmids (3rd generation lentivirus psPAX2 [viral polymerase)] and pMD2.g [envelope]) were a gift from Dr. Li Qian (University of North Carolina). HEK293T cells were transfected with the *ESR1*, *ESR2*, or mCherry plasmids, along with lentiviral plasmids to facilitate lentiviral packaging, as described [48]. HUVEC and HDMVEC were transduced with a dose-response of packaged lentiviral vector for 72 hours to determine the appropriate transduction dose. For lentiviral expression experiments, HUVEC and HDMVEC were transduced with 2 µL of lentiviral particles per 100,000 cells at day 0, switched to CSS media at day 2, treated with EE at day 3, then harvested and/or assayed at day 4.

### Statistical methods

Statistical analyses were performed in GraphPad Prism version 9. RT-qPCR relative expression values were $\log_2$-transformed and compared by one-way ANOVA and Šídák multiple comparison testing. Thrombin generation and fibrin formation parameters were compared by one-way ANOVA and Dunnett multiple comparison testing. For experiments involving lentiviral transduction and hormone treatment, significant differences were identified by two-way ANOVA and Šídák multiple comparison testing.

## Results

### OC hormones do not enhance EC-supported thrombin generation

To characterize the effect of OC hormones on ECs, we first exposed HUVEC and HDMVEC to concentrations of EE and/or drospirenone achieved in plasma during OC administration (1

nM and 100 nM, respectively) [49]. Transcriptional regulation is a primary effect of steroid hormones. To confirm OC hormone transcriptional activity *in vitro*, we treated MCF-7 breast epithelial cells with hormone preparations (EE and/or drospirenone) and confirmed each preparation upregulated progesterone receptor (*PGR*) transcription 5-10-fold, as expected (S1 Fig in S1 File) [50, 51]. Additional control experiments in HUVEC and HDMVEC showed 24-hour exposure to the inflammatory cytokine TNFα significantly reduced *TFPI* and *THBD* transcripts in both primary ECs, and elevated *F3* transcripts slightly, but non-significantly (HUVEC) or significantly (HDMVEC), confirming both cells had intact pro-inflammatory signaling (Fig 1A–1C). TNFα was subsequently used as a positive control to show the potential of primary ECs to become activated *in vitro*. However, neither EE nor drospirenone, alone or together, altered expression of *TFPI*, *THBD*, or *F3*, by HUVEC or HDMVEC (Fig 1A–1C).

Steroid hormones also have non-transcriptional effects, through which OCs may elicit non-genomic changes in EC functions [52–54]. To assess potential non-genomic effects on EC pro-coagulant activity, we measured the effect of EE and drospirenone on EC ability to support thrombin generation in NPP. Control experiments confirmed that in the absence of cells, neither EE nor drospirenone directly altered plasma thrombin generation in either the absence or presence of thrombomodulin (S2A, S2B Fig in S2 File). Untreated HUVEC and HDMVEC were relatively quiescent, although untreated HDMVEC showed slightly higher basal procoagulant activity than untreated HUVEC (Fig 1D and 1E). As expected, TNFα enhanced thrombin generation in both HUVEC and HDMVEC. However, neither EE nor drospirenone increased thrombin generation across a 3 log-fold dilution series (Fig 1D, 1E in S2 File) or for an extended period of time (5 days) in either cell type. We also triggered reactions in the presence of exogenous tissue factor to synchronize coagulation initiation and identify potential differences in the propagation phase; however, compared to vehicle-treated ECs, neither EE nor drospirenone increased thrombin generation (Fig 1F, 1G in S2 File). Hormone treatment also did not enhance thrombin generation in the presence of platelets. Together these results suggest neither EE nor drospirenone, alone or together, directly enhance EC procoagulant activity.

## OC hormones do not enhance EC-supported fibrin network formation

HUVEC and HDMVEC may modify fibrin networks independent of their ability to support thrombin generation, via interactions between fibrin and integrin $\alpha_v\beta_3$ [41]. ECs can also regulate fibrinolysis via expression of tissue plasminogen activator and PAI-1 [38–41]. To assess whether EE and/or drospirenone modify endothelial expression of these fibrin-modifying proteins, we treated HUVEC and HDMVEC with EE, drospirenone, or TNFα for 24 hours and measured gene expression by RT-qPCR. In both HUVEC and HDMVEC, TNFα significantly upregulated both *ITGAV* and *ITGB3* transcripts (Fig 2A and 2B). TNFα also upregulated *SERPINE1* in HUVEC (Fig 2C) and downregulated *PLAT* in HDMVEC (Fig 2D). The combination of EE and drospirenone slightly, but significantly, enhanced *PLAT* transcripts in HUVEC (Fig 2D); however, apart from this relatively small change, neither EE nor drospirenone, alone or together, altered expression of these fibrin network modifiers. In a parallel, independent experiment, we exposed a second lot of HDMVEC to EE and drospirenone and measured a panel of 84 endothelial-specific genes; this experiment confirmed that neither EE nor drospirenone significantly altered transcription of any of the tested genes (S3 Fig in S1 File).

We then tested whether EE and drospirenone can alter the ability of ECs to support fibrin formation via non-transcriptional mechanisms by treating cells with these hormones alone or together and measuring changes in turbidity, a marker of fibrin formation and structure [55]. Control experiments showed neither EE nor drospirenone altered fibrin formation in NPP in

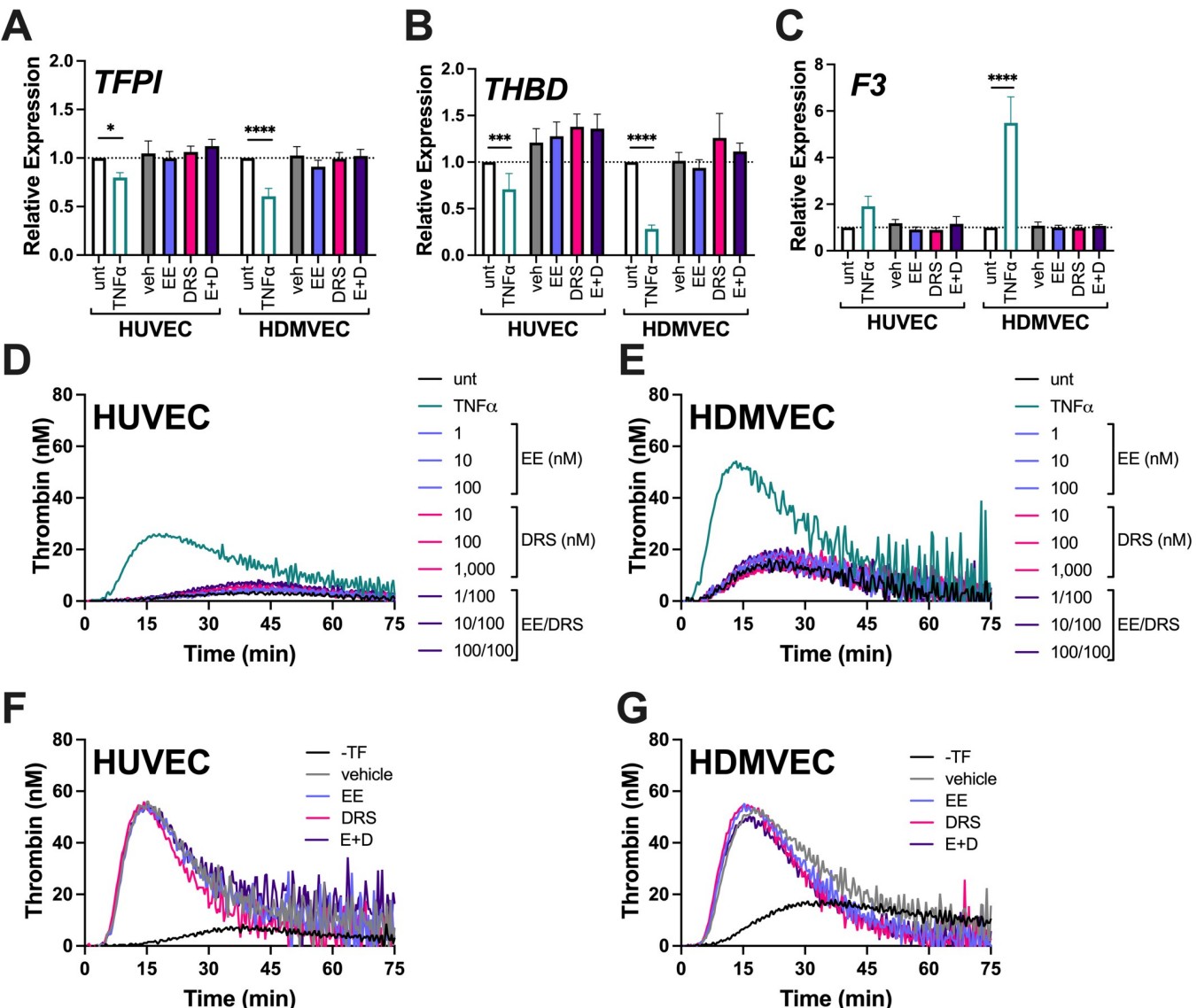

**Fig 1. OC hormones do not enhance EC-supported thrombin generation.** HUVEC and HDMVEC were treated with vehicle (0.7% ethanol [veh]), 10 ng/mL TNFα, 1 nM EE, 100 nM drospirenone (DRS), or EE and drospirenone (E+D) for 24 hours before RNA was extracted. Transcripts encoding **(A)** tissue factor pathway inhibitor (*TFPI*), **(B)** thrombomodulin (*THBD*), and **(C)** tissue factor (*F3*) were measured by RT-qPCR and normalized to the untreated control (N = 3–12; Bars = mean + SEM; *p<0.05, ***p<0.001, ****p<0.0001). **(D-G)** Thrombin generation was initiated in recalcified NPP by **(D)** HUVEC and **(E)** HDMVEC after 24 hours of TNFα, or EE and/or drospirenone. Curves are representative of N = 7 experiments performed in duplicate. Thrombin generation initiated by exogenous tissue factor and propagated by HUVEC **(F)** or HDMVEC **(G)** was measured in NPP (N = 3). Thrombin generation parameters and statistical analyses are provided in S2 File.

the absence of cells (S2C Fig in S2 File). As expected, TNFα treatment of either HUVEC or HDMVEC shortened the lag time and increased the rate of fibrin formation. OC-treated HUVEC showed a significant, though inconsistent, shortening of the time to plateau after some treatments compared to untreated ECs (S2 File); however, neither EE nor drospirenone enhanced HUVEC- or HDMVEC-mediated fibrin formation lag time, rate, or turbidity change (Fig 2E, 2F in S2 File). Moreover, the final turbidity of clots produced by these cells was stable, suggesting fibrin was not lysed following its formation in any of the conditions tested. Collectively, these data suggest that neither EE nor drospirenone enhance EC-supported clot formation.

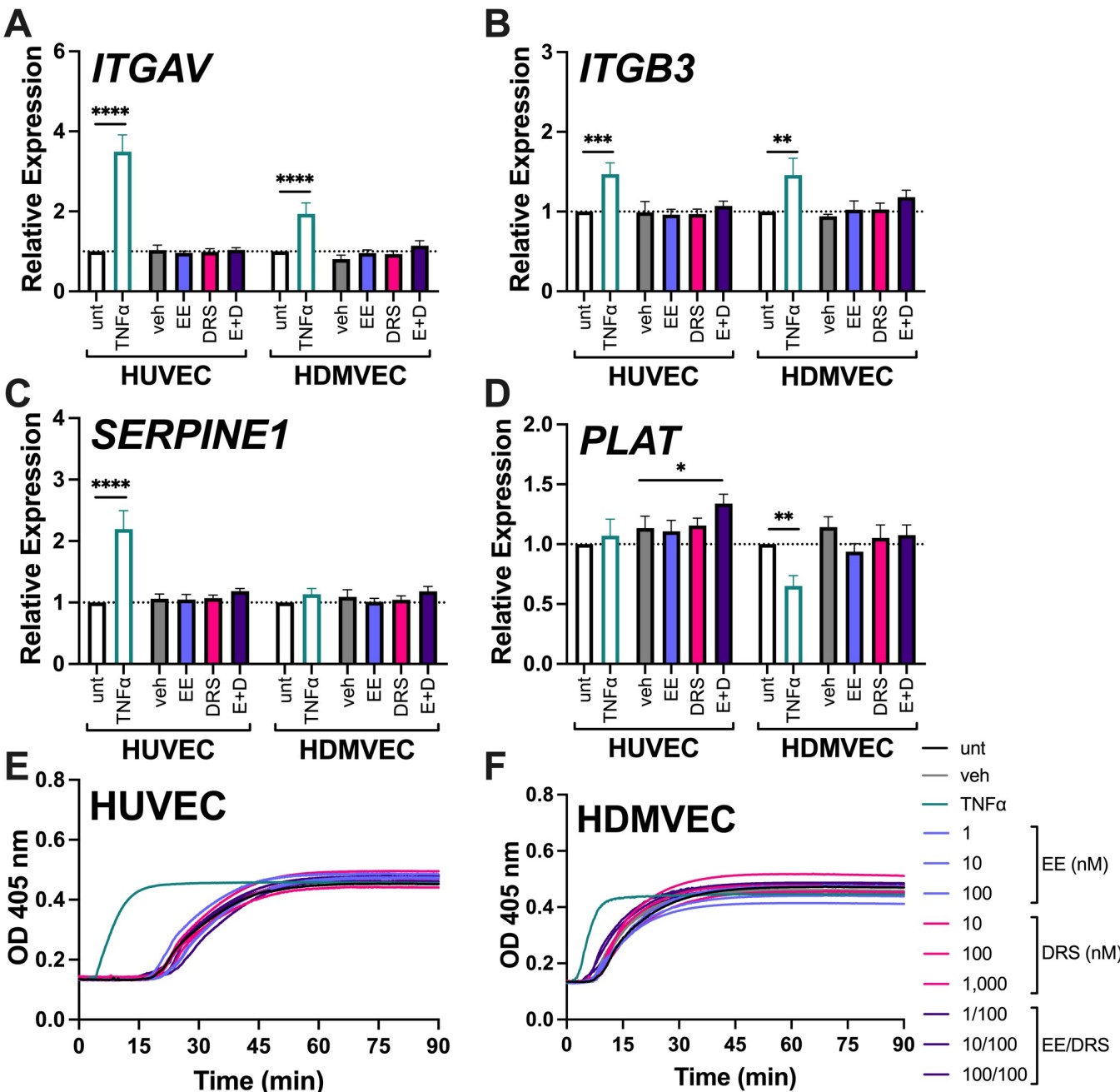

**Fig 2. OC hormones do not enhance EC-supported fibrin formation.** HUVEC and HDMVEC were treated with vehicle (0.7% ethanol [veh]), 10 ng/mL TNFα, 1 nM EE, 100 nM drospirenone (DRS), or EE and drospirenone (E+D) for 24 hours before RNA was extracted. Transcripts encoding **(A)** integrin αv (*ITGAV*), **(B)** integrin β3 (*ITGB3*), **(C)** plasminogen activator inhibitor type I (*SERPINE1*), and **(D)** tissue plasminogen activator (*PLAT*) were measured by RT-qPCR and normalized to the untreated control (N = 3–12; Bars = mean + SEM). TNFα was compared to untreated cells, whereas EE and drospirenone treatments were compared to vehicle (*p<0.05, ***p<0.001, ****p<0.0001). After 24 hours of TNFα, EE, and/or drospirenone treatment, fibrin formation initiated by **(E)** HUVEC and **(F)** HDMVEC was measured by turbidity in recalcified NPP (curves representative of N = 4–5). Fibrin formation parameters and statistical analyses are provided in S2 File.

## Primary ECs have low, but inter-individually variable, estrogen receptor expression

Effects of EE on cellular function are thought to occur via the estrogen receptors ERα and ERβ, encoded by the *ESR1* and *ESR2* genes, respectively [28, 32]. Both ERα and ERβ are nuclear hormone receptors that reside in the cytoplasm until ligand binding induces receptor translocation to the nucleus and localization to their transcriptional targets [56]. To determine if estrogen receptors are expressed in ECs, we mined gene expression data from a publicly available database of 53 human aortic EC lines [46]. This approach enabled us to probe the endothelial transcriptome in a specific vascular bed, albeit arterial instead of venous, across a population. We verified EC identity by high VWF expression (Fig 3A). Of the coagulation- and fibrin-related transcripts analyzed, *SERPINE1* was most abundant and *F3* was least abundant (Fig 3A). Interestingly, although *ESR1* and *ESR2* were only expressed at low levels in most

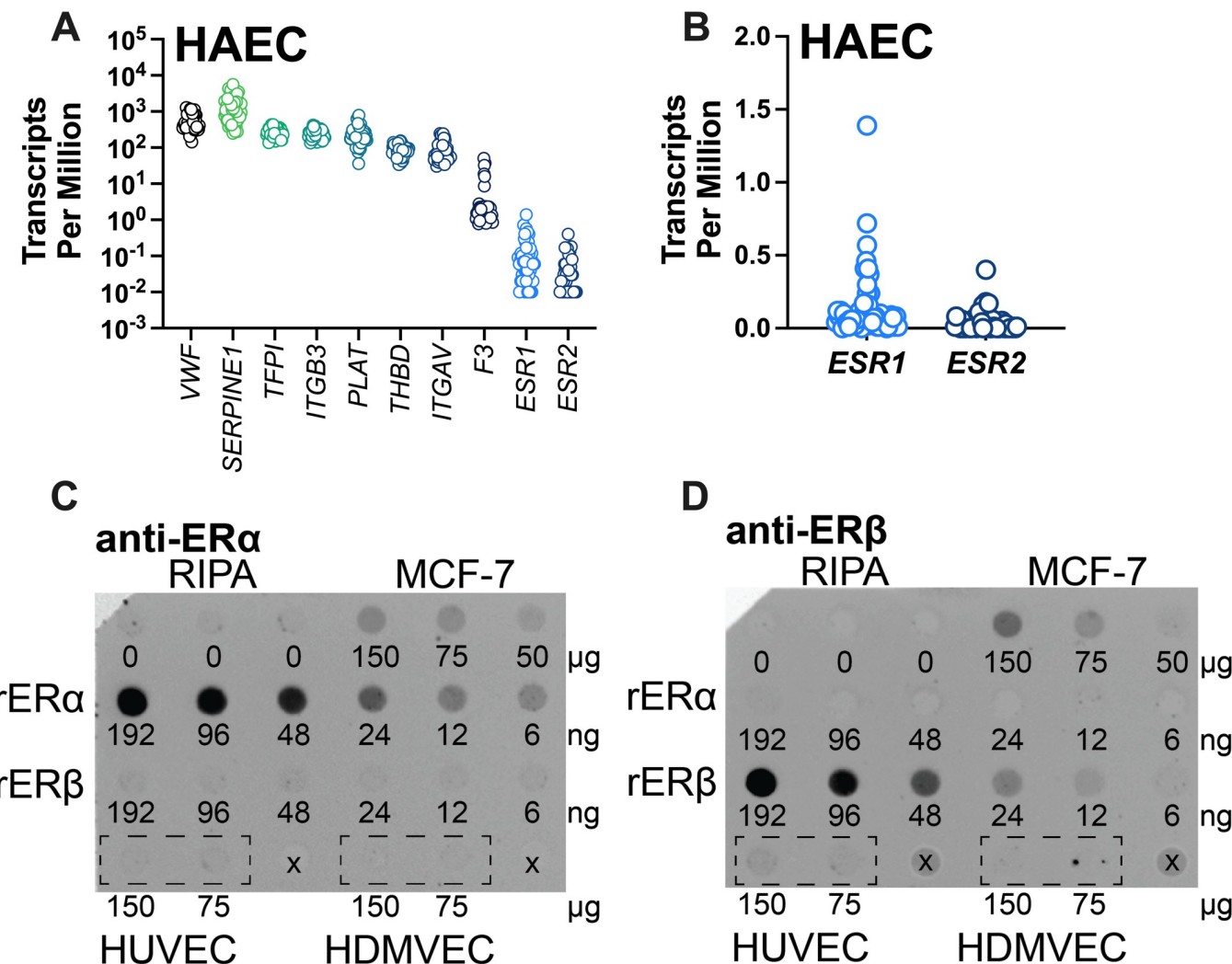

**Fig 3. Primary ECs have low, variable levels of estrogen receptor gene expression.** (**A**) RNA-seq data from aortic EC were mined for coagulation genes of interest. (**B**) Estrogen receptor gene expression shown on a linear scale. HUVEC and HDMVEC whole cell lysates were probed for (**C**) ERα and (**D**) ERβ protein, using MCF-7 cell lysates and recombinant ERα and ERβ as positive controls. Empty wells are marked with an x. Blot is representative of N = 3.

individuals, a subset of individuals had measurable expression of *ESR1* and *ESR2* transcripts (Fig 3B).

In contrast, analysis of our HUVEC and HDMVEC by RT-qPCR suggested low/undetectable or no expression of *ESR1* or *ESR2* ($C_T$>28), and immunoblots of cell lysates probed for ERα and ERβ proteins confirmed these receptors were below the level of detection (Fig 3C and 3D). Previous studies have shown that vascular endothelial cells do not express progesterone receptors [57], and we also confirmed that *PGR* transcript was absent in our cells ($C_T$>35). This analysis suggested expression of these key hormone receptors in ECs varies between individuals, raising the possibility that presence of *ESR1* and/or *ESR2* are needed to potentiate prothrombotic effects of OC hormones in certain individuals.

## Overexpression of *ESR1* or *ESR2* does not facilitate HUVEC or HDMVEC response to EE

To determine if estrogen receptor expression could enable ECs to respond to EE, we overexpressed *ESR1* (lenti-*ESR1*) and *ESR2* (lenti-*ESR2*), alone or together. Control experiments showed lenti-mCherry transduction induced a dose-dependent increase in fluorescence, indicating the ECs tolerated and responded to lentiviral transduction (Fig 4A, S4A Fig in S1 File). Subsequent dose-response experiments identified an optimal dose of viral particles for these studies (S4B Fig in S1 File). Analysis of HUVEC and HDMVEC transduced with lenti-mCherry control, -*ESR1*, -*ESR2*, or both -*ESR1+2* (half dose of each virus) showed both lentiviral constructs upregulated the corresponding transcript 5-15-fold over basal levels (Fig 4B and 4C). Following transduction of *ESR1* and *ESR2*, ERα protein was detectable at levels ~8-fold higher than MCF-7 cells, but ERβ protein was below the level of detection (Fig 4D and 4E). Cells transduced with both constructs were used in subsequent experiments to test hormone responses.

To then determine whether expression of *ESR1* and/or *ESR2* potentiates a transcriptional response to EE, we transduced HUVEC and HDMVEC with lenti-mCherry, -*ESR1*, and/or -*ESR2* for 72 hours, exposed the cells to 1 nM EE for 24 hours, and characterized gene expression and procoagulant activity. Relative expression values were compared to the lenti-mCherry control in each treatment group. Exposure to EE did not significantly change the level of lentiviral-induced *ESR1* or *ESR2* transcription in HUVEC or HDMVEC. Neither ER expression nor exposure to EE significantly altered *TFPI*, *THBD*, or *F3* transcripts (Fig 5A–5C). Overexpression of either receptor reduced *ITGAV* transcripts independent of EE exposure non-significantly in HUVEC and significantly in HDMVEC (Fig 5D), but did not reduce *ITGB3* expression (Fig 5E). *SERPINE1* was slightly, but consistently reduced after *ESR2* overexpression, independent of EE (Fig 5F), but *PLAT* was unchanged (Fig 5G). Importantly, despite these small changes, subsequent analysis showed that EC ability to support thrombin generation was not enhanced by overexpression of *ESR1* or *ESR2* or EE exposure in either HUVEC or HDMVEC (Fig 5H, 5I in S2 File).

## Inflammation does not potentiate EE-mediated effects on HUVEC and HDMVEC

Finally, we tested the hypothesis that a combination of two "hits"—exposure to exogenous hormones plus inflammation—promotes EC prothrombotic activity during OC use. We first confirmed that TNFα did not alter transcription of *ESR1* or *ESR2* in lentiviral-transduced ECs (S5A, S5B Fig in S1 File). We then transduced HUVEC and HDMVEC with lentiviral constructs for 72 hours, treated cells with 1 nM EE alone or in combination with 10 ng/mL TNFα for 24 hours, and analyzed gene expression and thrombin generation. RT-qPCR data were

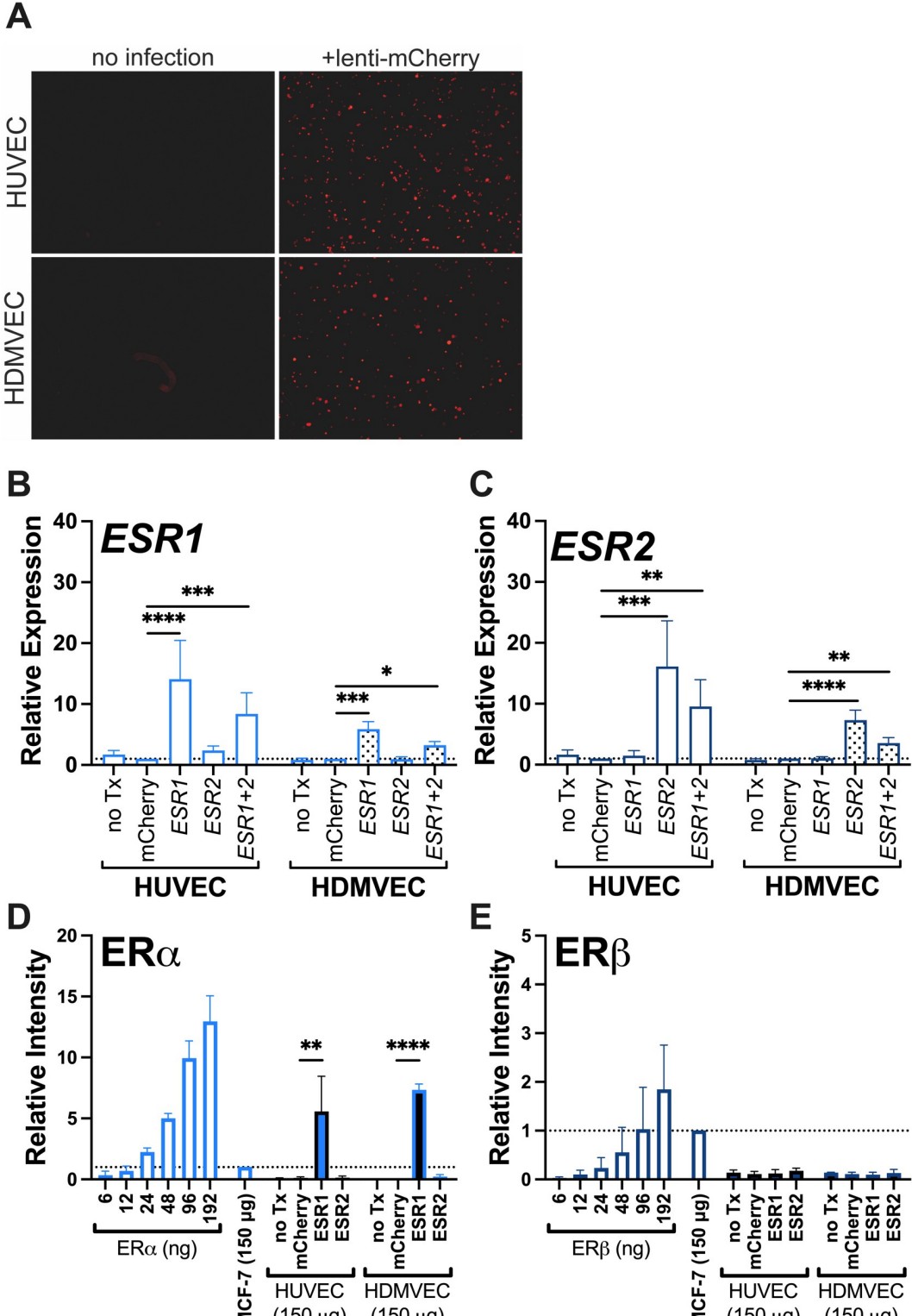

**Fig 4. Lentiviral-mediated overexpression of *ESR1* and *ESR2*. (A)** HUVEC and HDMVEC tolerance of lentiviral transduction was measured by fluorescence 72 hours after lenti-mCherry transduction. **(B)** *ESR1* and **(C)** *ESR2* expression was measured by RT-qPCR and normalized to lenti-mCherry samples (N = 4). Relative expression of **(D)** ERα and **(E)** ERβ protein in HUVEC and HDMVEC following lentiviral transduction was normalized using MCF-7 immuno-dot blot intensity (N = 3). Bars = mean + SEM; *p<0.05, **p<0.01, ***p<0.001, ****p<0.0001.

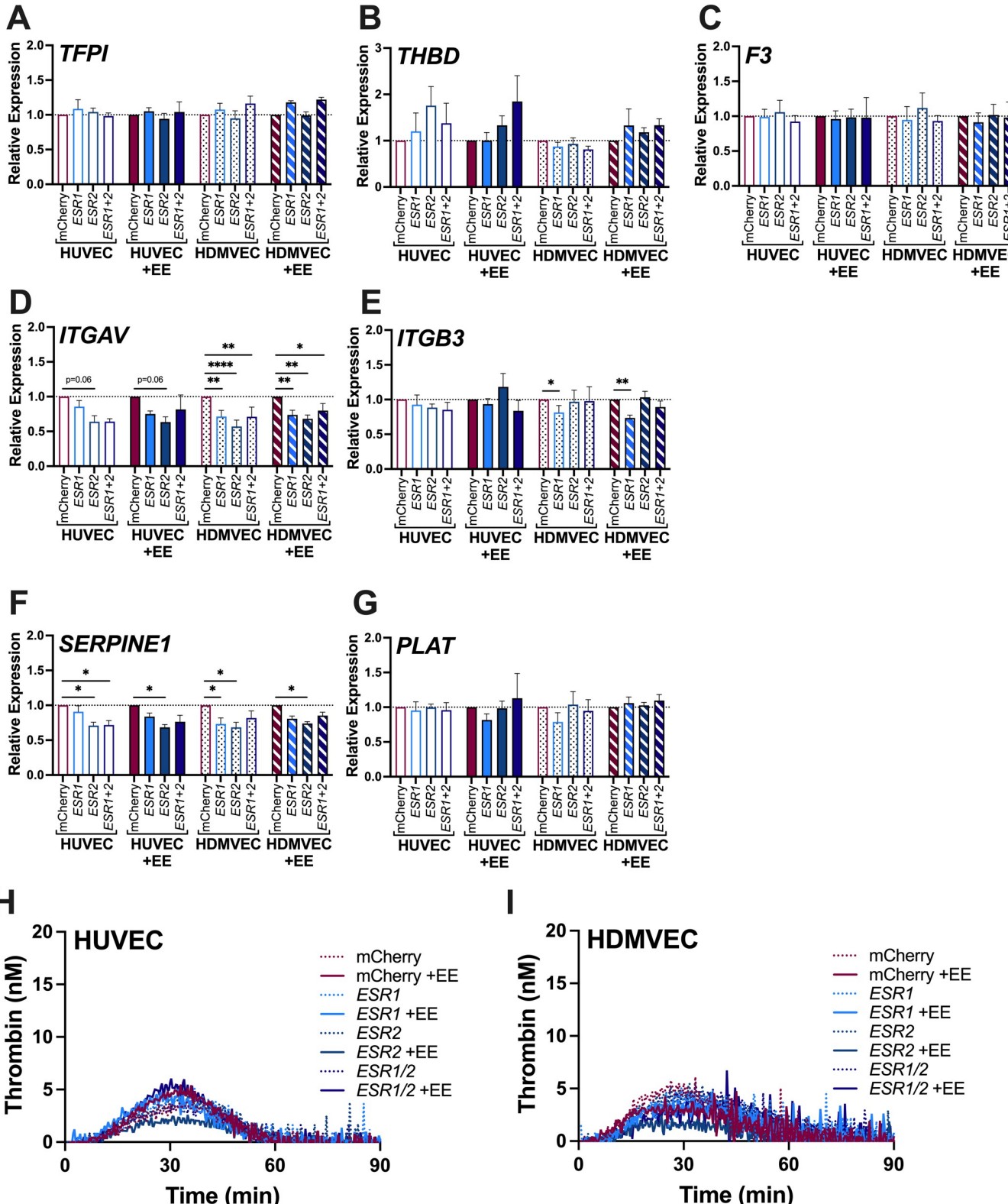

**Fig 5. Overexpression of *ESR1* and *ESR2* in HUVEC and HDMVEC does not facilitate a prothrombotic response to EE.** Transcript expression of (**A**) *TFPI*, (**B**) *THBD*, (**C**) *F3*, (**D**) *ITGAV*, (**E**) *ITGB3*, (**F**) *SERPINE1*, and (**G**) *PLAT* was measured by RT-qPCR following 72 hours lentiviral transduction and 24 hours of subsequent EE treatment (N = 4; Bars = mean + SEM; *p<0.05, **p<0.01, ***p<0.001, ****p<0.0001). Thrombin generation in NPP stimulated by (**H**) HUVEC or (**I**) HDMVEC treated with lentiviral vector and EE (single curves representative of N = 4 experiments in duplicate). Thrombin generation parameters and statistical analyses are provided in S2 File.

normalized to lenti-*ESR1+2*-infected HUVEC or HDMVEC, which did not have altered expression compared to untreated cells. Compared to TNFα treatment, which significantly downregulated *TFPI* and *THBD* in *ESR*-transfected HDMVEC, TNFα plus EE treatment slightly increased transcription of both anticoagulants (Fig 6A and 6B), suggesting a potentially and paradoxically protective effect. The TNFα-induced changes in expression of *F3* and *ITGAV* were not modified by co-exposure to EE (Fig 6C and 6D). TNFα-induced increases in *ITGB3* and *SERPINE1* expression were mildly enhanced by EE (Fig 6E and 6F). Cotreatment with EE did not change the decreased *PLAT* expression caused by TNFα in HDMVEC (Fig 6G). In thrombin generation assays, neither lentiviral overexpression of estrogen receptors nor EE treatment modified the EC procoagulant response to TNFα (Fig 6H, 6I in S2 File). Altogether, we were unable to observe any tractable hormone-induced changes in endothelial procoagulant function, suggesting the minor transcriptional changes we detected were not sufficient to produce measurable differences in procoagulant activity.

## Discussion

Throughout history, advances in contraceptive measures such as OCs have been imperative to women's healthcare, empowerment, and global economic development [1]. Not only does contraception prevent unwanted pregnancies, but it also reduces maternal mortality and the need for more aggressive methods of reproductive healthcare including abortions, which are illegal, difficult, and unsafe to access in many countries. To preserve women's healthcare and autonomy, it is essential to identify mechanisms that associate OC use with VTE to understand how to provide safe and effective OCs. Although negative, our study advances the field via its essential, fundamental observations. Our data rigorously show that direct exposure to high-risk OCs does not provoke EC procoagulant activity. Moreover, expression of genes encoding the canonical estrogen receptors is not sufficient to "unlock" EC responses to EE, even in the presence of an inflammatory stimulus. These observations refute a mechanism in which OCs promote VTE by directly enhancing EC procoagulant activity, but raise alternative hypotheses. These include the potential that hormone-responsive cells promote secondary endothelial dysfunction in OC users, and/or that individual heterogeneity in steroid signaling or metabolism promotes VTE in a small proportion of OC users. Ultimately our findings suggest future efforts to understand the link between OC use and VTE should be directed towards defining potential interactive and/or synergistic effects of OC hormones and individually programmed responses to hormones in the presence of additional VTE risk factors.

Whereas 17β-estradiol, the endogenously produced estrogen, promotes HUVEC proliferation and regulates metabolism and vasodilation gene expression via estrogen receptors [58, 59], our findings that EE and drospirenone do not directly change EC procoagulant activity are consistent with prior observations in HUVEC that EE does not increase expression of tissue factor or downregulate thrombomodulin activity [60]. Additional reports have shown only minimal effects of EE on other HUVEC functions, including synthesis of DNA, nitric oxide, prostaglandin, and prostacyclin [61–65]. However, since endothelial dysfunction is thought to initiate thrombus formation, women who develop VTE may carry undiscovered risk factors or polymorphisms that interact with OCs to facilitate this transition in venous endothelium. Notably, our findings do not rule out an indirect role of endothelium or endothelial activation in OC-related VTE. For example, OC hormones may activate hepatocytes, other blood cells (e.g., platelets, neutrophils, monocytes) or hormone-responsive tissues (uterine, ovarian, breast) that secondarily release factors that activate ECs and spark procoagulant activity at the vessel wall.

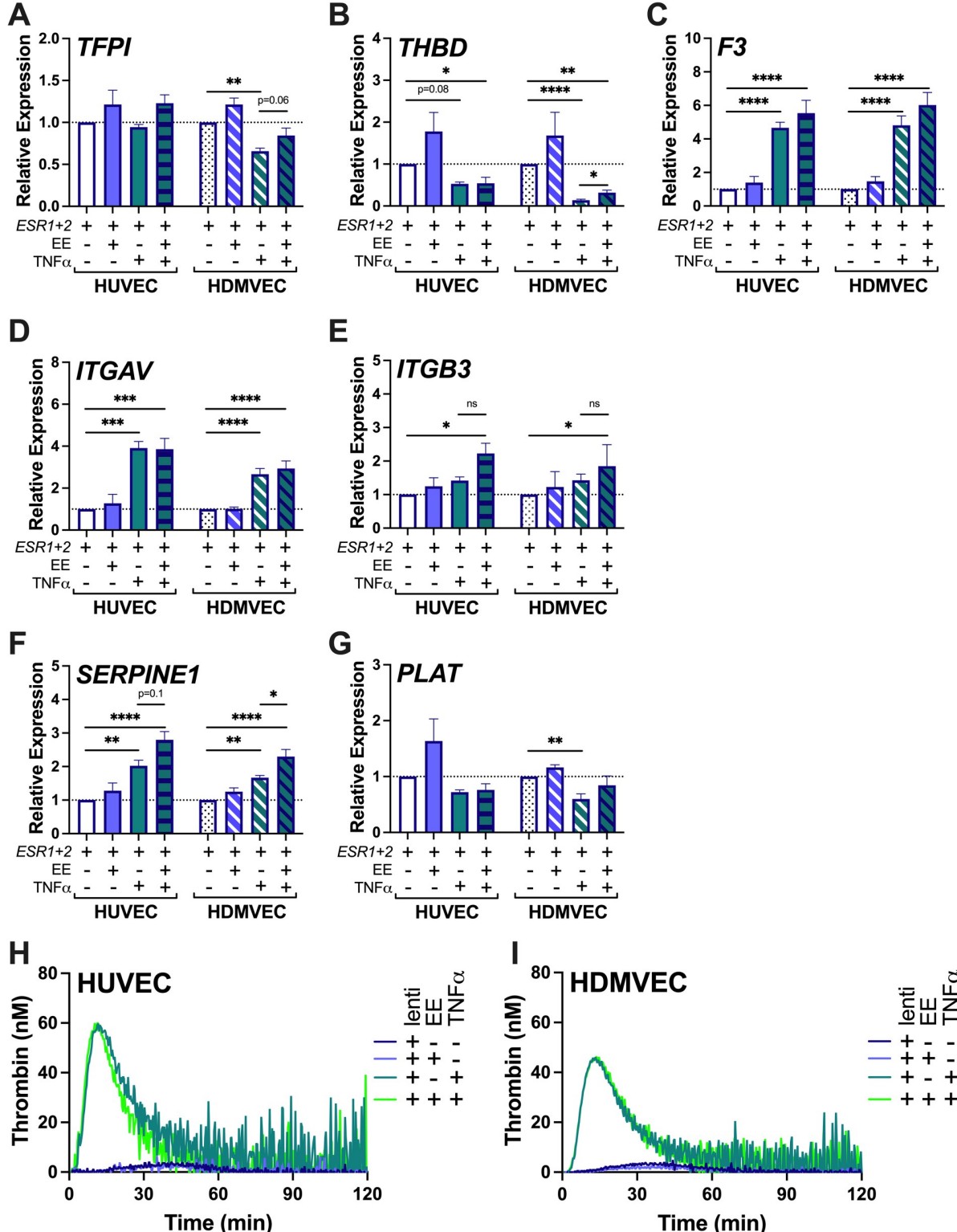

**Fig 6. Combined inflammation and overexpression of *ESR1* and *ESR2* in HUVEC and HDMVEC does not facilitate a prothrombotic response to EE.** Transcript expression of (**A**) *TFPI*, (**B**) *THBD*, (**C**) *F3*, (**D**) *ITGAV*, (**E**) *ITGB3*, (**F**) *SERPINE1*, and (**G**) *PLAT* was measured by RT-qPCR following 72 hours lentiviral transduction and 24 hours of subsequent 1 nM EE and/or 10 ng/mL TNFα treatment (N = 4; Bars = mean + SEM; *p<0.05, **p<0.01, ***p<0.001, ****p<0.0001). The lenti-ESR1+2, and lenti-ESR1+2 +EE data are reproduced from Fig 5 for comparison. Thrombin generation in NPP stimulated by (**H**) HUVEC or (**I**) HDMVEC treated with lentiviral vector and EE (single curves representative of N = 4 experiments in duplicate). Thrombin generation parameters and statistical analyses are provided in S2 File.

An interesting, but perplexing finding from our study is that even overexpression of genes encoding the canonical estrogen receptors did not "unlock" responses of ECs to EE exposure. Estrogen signaling is regulated by chromatin structure at estrogen-responsive elements, expression of transcriptional co-activators and/or repressors, and receptor conformational states [66, 67]. Thus, signaling through the steroid receptors may require additional genomic or cellular machinery that is not present in ECs. Accordingly, while negative, our findings that the EE-ER signaling axis does not promote procoagulant activity in primary ECs opens avenues for future investigation. First, nuclear estrogen receptors belong to a large, promiscuous family of steroid receptors. The synthetic steroid-like hormones present in OC preparations may interact with receptors other than the canonical estrogen and progesterone receptors. For example, unlike most progestins which are derived from the natural hormone progesterone, drospirenone is derived from the synthetic variant spironolactone, and may act via androgen and mineralocorticoid nuclear receptors [68–71]. EE may also act through other estrogen receptors, such as the G protein-coupled estrogen receptor, which can initiate intracellular signaling cascades including nitric oxide pathways [54]. Second, the hormone preparations tested here were able to induce *PGR* expression in MCF-7 cells, indicating these preparations are biologically active and have direct signaling potential. However, hormones administered orally undergo hepatic bypass before circulating through the body. Thus, secondary metabolites (e.g., estradiol sulfates) may impact EC procoagulant activity through direct or indirect mechanisms [72, 73]. Interactions between other potentially active receptors and steroids molecules can be investigated using our *in vitro* design in future studies.

Our experimental model enabled us to explore the hypothesis that OC-related VTE involves multiple hits to the endothelium that provoke thrombus formation. Although our data suggest EE does not interact with TNFα-induced inflammation, other inflammatory events may fill this role. For instance, environmental risk factors such as obesity-related metabolic disorders or smoking may induce underlying endothelial dysfunction that is exacerbated by OC use through unrecognized mechanisms [74–77]. Alternatively, plasma thrombophilias may interact with the endothelium during OC use [78, 79]. Khialani et al observed that presence of FV Leiden modestly increases risk of OC-related VTE [80], and Vandenbroucke et al found women with combined FV Leiden heterozygosity and OC use had a 30-fold higher risk of VTE than women without either risk factor [81]. Our experimental model of the EC/plasma interface can be used to investigate interactions between OC hormones and other known VTE risk factors such as FV Leiden in future studies. In addition, our experimental platform could be used to probe for factors in plasma from OC users that may impact the endothelium.

This study had limitations. First, although VTE most commonly occurs in large veins, deep vein ECs from premenopausal, female donors were not commercially available. However, we tested two primary EC types in the current study–HUVEC from a large vessel, albeit with significant previous hormone exposure, and HDMVEC which are microvascular in origin but were harvested from two separate, pre-menopausal, female donors–and both EC types showed similar responses throughout our study. Second, since currently there are no publicly available datasets detailing venous endothelial gene expression, we were only able to analyze gene expression data from primary aortic ECs; however, observing ER gene expression in a subset of these cells was sufficient to motivate subsequent experiments in venous ECs. Third, although ECs are exposed to flow *in vivo*, we did not culture HUVEC or HDMVEC under flow. However, static culture conditions may recapitulate blood stasis, an established element of Virchow's triad. Similarly, hypoxic conditions may promote thrombus development. Many women form OC-related VTE in the absence of identifiable provoking factors. Future studies should investigate whether hypoxia combined with hormone treatment enhances procoagulant activity. Finally, although our study comprehensively assessed gene expression and

prothrombotic potential after directly exposing ECs to OCs, additional genetic, cellular, and pharmacologic variables may mediate endothelial dysregulation in OC-related VTE. Additional studies are needed to identify these potential pathways and disentangle mechanisms leading to OC-related VTE.

In summary, we developed an *in vitro* model of the endothelial-blood interface to study primary, direct effects of high-risk hormones on endothelial cells. We found that EE and drospirenone do not directly provoke EC procoagulant activity, and over-expression of canonical nuclear hormone receptors led to only minimal transcriptional effects that were inconsistent with prothrombotic functions. Although these data do not support a mechanism in which exogenous hormones in OC formulations directly induce EC procoagulant activity, they do not rule out an indirect role of the endothelium in OC-related VTE. Establishing the current observations was a necessary and important step towards deducing pathophysiologic mechanisms mediating OC-related VTE.

## Supporting information

**S1 File. Contains supporting figures and tables.**
(DOCX)

**S2 File. An Microsoft excel document containing quantitative parameters from thrombin generation and fibrin formation experiments.**
(XLSX)

**S3 File. Holds the raw dot blot images underlying Figs 3 and 4.**
(PDF)

## Acknowledgments

We acknowledge the complexity of sex and gender in society and medicine. We used genetically female cells for this study, and although we referenced "women" as the relevant population throughout the text, we appreciate that not all people who use oral contraceptives self-identify as women. We also thank Lori A. Holle, Dre'Von A. Dobson, Benjamin Keepers, and Yi Yang for their technical assistance and advice.

## Author Contributions

**Conceptualization:** Emma G. Bouck, Nicholas L. Smith, Charles J. Lowenstein, Alisa S. Wolberg.

**Data curation:** Emma G. Bouck, William O. Osburn, Yaqiu Sang.

**Formal analysis:** Emma G. Bouck, Marios Arvanitis, William O. Osburn, Alisa S. Wolberg.

**Funding acquisition:** Nicholas L. Smith, Charles J. Lowenstein, Alisa S. Wolberg.

**Investigation:** Emma G. Bouck, Paula Reventun.

**Methodology:** Emma G. Bouck, Marios Arvanitis, Yaqiu Sang.

**Supervision:** Homa K. Ahmadzia, Alisa S. Wolberg.

**Visualization:** Emma G. Bouck.

**Writing – original draft:** Emma G. Bouck.

**Writing – review & editing:** Emma G. Bouck, Marios Arvanitis, William O. Osburn, Yaqiu Sang, Paula Reventun, Homa K. Ahmadzia, Nicholas L. Smith, Charles J. Lowenstein, Alisa S. Wolberg.

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
