## [Decision Letter · Decision Letter 0]

8 Mar 2023

PONE-D-22-34362High risk oral contraceptive hormones do not directly enhance endothelial cell procoagulant activityPLOS ONE

Dear Dr. Wolberg:

Thank you for submitting your manuscript to PLOS ONE. After careful consideration, we feel that it has merit but does not fully meet PLOS ONE’s publication criteria as it currently stands. Therefore, we invite you to submit a revised version of the manuscript that addresses the points raised during the review process.

We look forward to receiving your revised manuscript.

Kind regards,

James Peter Maloney, MD

Academic Editor

PLOS ONE

Journal Requirements:

In your cover letter, please note whether your blot/gel image data are in Supporting Information or posted at a public data repository, provide the repository URL if relevant, and provide specific details as to which raw blot/gel images, if any, are not available. Email us at plosone@plos.org if you have any questions

Additional Editor Comments (if provided):

Thank you for submission of this very interesting study. Please address the comments of the two peer reviewers. A repeat of the experiments using plasma (added to culture media) from donors on OCPs is interesting and needs to be discussed, but is likely beyond the scope of the project unless the authors already have such plasma frozen away and ready for experiments.

Reviewers' comments:

Reviewer's Responses to Questions

**Comments to the Author**

1. Is the manuscript technically sound, and do the data support the conclusions?

Reviewer #1: Yes

Reviewer #2: Partly

2. Has the statistical analysis been performed appropriately and rigorously? 

Reviewer #1: Yes

Reviewer #2: Yes

3. Have the authors made all data underlying the findings in their manuscript fully available?

Reviewer #1: Yes

Reviewer #2: Yes

4. Is the manuscript presented in an intelligible fashion and written in standard English?

Reviewer #1: Yes

Reviewer #2: Yes

5. Review Comments to the Author

Reviewer #1: In this study, Bouck et al. examine the possibility that the contraceptive hormones esthinyl estradiol and drospirenone are transcriptional regulators of a prothrombotic endothelial phenotype. The authors should be congratulated on a very comprehensive body of work examining this issue. The mechanisms underlying contraceptive-related thrombotic risk have long been elusive. Thus, this negative study is important because it will help narrow the focus in ongoing investigations to identify alternative mechanistic explanations. The experiments examining the possibility of an inflammatory interaction with estrogen receptor signaling are particularly interesting. However, before fully discarding endothelial effects as an underlying mechanism, it may be worthwhile to examine some alternative hypotheses. The majority of contraceptive-related VTE present as unprovoked (other than the contraceptive), suggesting that inflammation, injury, and other provocations are unlikely to be involved. It is now thought that most unprovoked DVT arise from hypoxic deep vein valve pockets. It would thus be appropriate to determine if there is a relationship between contraception exposure and hypoxic endothelial response pathways (e.g. HIF1alpha). These pathways also mediate gene transactivation and may converge or synergize with estrogen receptor signaling to induce a prothrombotic phenotype. It would be especially interesting if HIF1alpha upregulates estrogen receptor expression.

Reviewer #2: Bouck and colleagues set out to study the effects of oral contraceptive use on the endothelial procoagulant activity in an in vitro setting. As the ECs used in these experiments have low/undetectable estrogen receptor levels (progesterone levels appear not to be measured?), it might not be that surprising that the data is predominantly negative. Also an attempt to overexpress the estrogen receptors followed by hormone treatment did not show significant effects on the procoagulant activity of ECs. While these in themselves are important observations, I do believe that the in vitro setting does not allow the strong conclusion drawn from these experiments as stated in the abstract that ‘the OC hormones EE and drospirenone do not directly induce prothrombotic activity in ECs’.

Major points:

1) Although EE/drospirenone does not affect expression of the genes of interest, cells were only incubated for 24 hours. Do the authors believe this would be sufficient time to see the expected changes in transcription and/or protein expression given the respective half times for these proteins?

2) Cells exposed to hormones were incubated with normal pooled plasma. However, in an in vivo setting, OC hormones would affect both the endothelium and levels of coagulation factors in the plasma. Performing a similar experiment using pooled plasma from OC users would strengthen the study.

3) The overall conclusions and discussion should more clearly reflect that these data were obtained from an in vitro system. Discussing available in vivo data would be helpful to see how these results compare.

Minor points:

- The rationale to use TNFa in itself is initially not clear; is it used as a positive control to show ECs in vitro can be activated? Or is EE/progestin believed to activate the endothelium? When discussing the two-hit hypothesis the use of TNFa is clearly justified.

- The methods section states the use of 17b-estradiol, but this is not mentioned in the results section? Where there any direct comparisons done between the EE and 17b-estradiol to determine if ECs in vitro are sensitive to naturally occurring estrogen?

- Was a dose-response curve performed to determine if higher hormone concentrations, potentially via indirect ways, do have an effect on the coagulation phenotype?

6. PLOS authors have the option to publish the peer review history of their article (what does this mean?). If published, this will include your full peer review and any attached files.

Reviewer #1: No

Reviewer #2: No

---

## [Author Response · Author response to Decision Letter 0]

16 Mar 2023

March 15, 2023

Dear Dr. Maloney and Reviewers:

Thank you for your review of our manuscript, “High risk oral contraceptive hormones do not directly enhance endothelial cell procoagulant activity in vitro” (PONE-D-22-34362). We appreciate the positive comments and thoughtful suggestions, and have revised our manuscript in response. Modified text is noted with bright blue font in the manuscript. 

Specific responses to the Reviewers’ comments are below:

Reviewer #1: 

In this study, Bouck et al. examine the possibility that the contraceptive hormones ethinyl estradiol and drospirenone are transcriptional regulators of a prothrombotic endothelial phenotype. The authors should be congratulated on a very comprehensive body of work examining this issue. The mechanisms underlying contraceptive-related thrombotic risk have long been elusive. Thus, this negative study is important because it will help narrow the focus in ongoing investigations to identify alternative mechanistic explanations. The experiments examining the possibility of an inflammatory interaction with estrogen receptor signaling are particularly interesting. However, before fully discarding endothelial effects as an underlying mechanism, it may be worthwhile to examine some alternative hypotheses. The majority of contraceptive-related VTE present as unprovoked (other than the contraceptive), suggesting that inflammation, injury, and other provocations are unlikely to be involved. It is now thought that most unprovoked DVT arise from hypoxic deep vein valve pockets. It would thus be appropriate to determine if there is a relationship between contraception exposure and hypoxic endothelial response pathways (e.g. HIF1alpha). These pathways also mediate gene transactivation and may converge or synergize with estrogen receptor signaling to induce a prothrombotic phenotype. It would be especially interesting if HIF1alpha upregulates estrogen receptor expression.

RESPONSE: We agree. We measured HIF1A transcripts as part of the 84-gene endothelial panel (Figure S3) but this did not change following hormone treatment (fold change=1.25; p=0.57). Therefore, we did not further probe hypoxic pathways, but our negative functional data suggest that even if these pathways are activated, they do not promote procoagulant activity in our system. Our findings could be expanded by using hypoxic chambers to evaluate the impact of hormone exposure during hypoxia. We have updated our limitations section to incorporate this point (Discussion, lines 415-417).

Reviewer #2: 

Bouck and colleagues set out to study the effects of oral contraceptive use on the endothelial procoagulant activity in an in vitro setting. As the ECs used in these experiments have low/undetectable estrogen receptor levels (progesterone levels appear not to be measured?), it might not be that surprising that the data is predominantly negative. Also an attempt to overexpress the estrogen receptors followed by hormone treatment did not show significant effects on the procoagulant activity of ECs. While these in themselves are important observations, I do believe that the in vitro setting does not allow the strong conclusion drawn from these experiments as stated in the abstract that ‘the OC hormones EE and drospirenone do not directly induce prothrombotic activity in ECs’.

RESPONSE: We agree. Previous studies have shown that vascular endothelial cells do not express progesterone receptors,[1] and we confirmed this in our endothelial cells (CT > 35). We have added a sentence to state this (lines 292-294). Although PR expression can be induced and mediate responses to progesterone treatment in ECs,[2] the progestin we used, drospirenone, is derived from spironolactone rather than progesterone. This steroid is thought to alternatively interact with mineralocorticoid and androgen receptors that are transcriptionally present in HUVEC and HDMVEC (Mean CT(NR3C2) = 25.9 and 26.0 respectively; Mean CT(AR) = 28.1 and 33.4, respectively) but did not facilitate a response. Thus, we focused on interactions between ethinyl estradiol and estrogen receptors in our study. 

To better temper the conclusions from our study, we have edited the title and abstract to more appropriately reflect the in vitro nature of our study.

Major points:

1) Although EE/drospirenone does not affect expression of the genes of interest, cells were only incubated for 24 hours. Do the authors believe this would be sufficient time to see the expected changes in transcription and/or protein expression given the respective half times for these proteins?

RESPONSE: In addition to the 24-hour experiments, we also incubated the ECs with EE/drospirenone for 5 days (replenishing the hormone-containing media each day) to test whether prolonged exposure led to functional changes in protein expression. Although prolonged incubation with the vehicle (0.7% ethanol) slightly enhanced thrombin generation, hormone treatment did not enhance cellular responses, even over this extended time course. These data are described in the results (line 244) and included in our data deposit file.

2) Cells exposed to hormones were incubated with normal pooled plasma. However, in an in vivo setting, OC hormones would affect both the endothelium and levels of coagulation factors in the plasma. Performing a similar experiment using pooled plasma from OC users would strengthen the study.

RESPONSE: We share the reviewer’s interest in expanding our endothelial platform to include plasmas isolated from OC users. Indeed, our findings have led to collaborative efforts to collect a cohort of appropriate plasmas for these studies. We are committed to these experiments but are undertaking these as a next-level study that is beyond the scope of the current manuscript. However, we have expanded the discussion to acknowledge these future studies (lines 403-404).

3) The overall conclusions and discussion should more clearly reflect that these data were obtained from an in vitro system. Discussing available in vivo data would be helpful to see how these results compare.

RESPONSE: We have edited the title and abstract to more appropriately reflect the in vitro nature of our study. Unexpectedly, rodent models of hormone-induced thrombosis have demonstrated antithrombotic effects of ethinyl estradiol and progestins.[3–9] These findings could be due to subcutaneous administration of the hormones rather than oral, or the concomitant ovariectomy that may have confounded the impact of exogenous hormones in a setting quite different than the human context.[3–5] Even in models utilizing oral gavage (still with ovariectomy), Cleuren et al found decreased in vivo thrombus weight, reduced plasma activity of procoagulant factors, and reduced ex vivo thrombin generation.[6–8] However, the reduced expression and activity of procoagulant proteins was dependent on estrogen receptor α;[4,8] this observation prompted us to investigate the estrogen receptors in vitro. We have added this point to our introduction to better frame the rationale for our in vitro study (lines 93-96).

Minor points:

- The rationale to use TNFa in itself is initially not clear; is it used as a positive control to show ECs in vitro can be activated? Or is EE/progestin believed to activate the endothelium? When discussing the two-hit hypothesis the use of TNFa is clearly justified.

RESPONSE: Indeed, we used TNFα first as a positive control and later as a potential contributor in a “2-hit” hypothesis. We have updated the first section in our results (line 232-233) to better introduce the rationale for using TNFα in these experiments.

- The methods section states the use of 17b-estradiol, but this is not mentioned in the results section? Where there any direct comparisons done between the EE and 17b-estradiol to determine if ECs in vitro are sensitive to naturally occurring estrogen?

RESPONSE: We apologize for this error. Other groups previously showed that 17β-estradiol regulates proliferation and regulates metabolism and vasodilation gene expression via estrogen receptors in HUVEC.[10,11] We did not directly compare EE and 17β-estradiol in this study. We edited the methods to correct this (line 141).

- Was a dose-response curve performed to determine if higher hormone concentrations, potentially via indirect ways, do have an effect on the coagulation phenotype?

RESPONSE: Since the circulating concentrations of ethinyl estradiol and drospirenone are believed to be ~1 nM and ~100 nM, respectively, we tested these and higher concentrations in a 3-fold log dose-response series. We observed that even 100X these concentrations did not influence the coagulation phenotype. These data are shown in figures 1 and 2.

We have also submitted a Supplemental Materials 3 pdf containing raw blot images underlying Figures 3 and 4.

We have also submitted a “final” version of the manuscript in which the figure legends are embedded within the manuscript text, as required by PLOS One.

We hope that with these modifications and explanations the manuscript is now acceptable for publication in PLOS One. Thank you for your consideration.

Sincerely, on behalf of the authors,

Emma Bouck

Alisa S. Wolberg, PhD

REFERENCES

1. Perrot-Applanat M, Cohen-Solal K, Milgrom E, Finet M. Progesterone receptor expression in human saphenous veins. Circulation. 1995;92: 2975–2983. doi:10.1161/01.CIR.92.10.2975

2. Goddard LM, Ton AN, Org T, Mikkola HKA, Iruela-Arispe ML. Selective suppression of endothelial cytokine production by progesterone receptor. Vascul Pharmacol. 2013;59: 36–43. doi:10.1016/j.vph.2013.06.001

3. Moverare S, Skrtic S, Lindberg MK, Dahlman-Wright K, Ohlsson C. Estrogen increases coagulation factor V mRNA levels via both estrogen receptor-alpha and -beta in murine bone marrow/bone. Eur J Endocrinol. 2004;151: 259. doi:10.1530/eje.0.1510259

4. Valéra MC, Gratacap MP, Gourdy P, Lenfant F, Cabou C, Toutain CE, et al. Chronic estradiol treatment reduces platelet responses and protects mice from thromboembolism through the hematopoietic estrogen receptor α. Blood. 2012;120: 1703–1712. doi:10.1182/blood-2012-01-405498

5. Valéra MC, Noirrit-Esclassan E, Dupuis M, Buscato M, Vinel A, Guillaume M, et al. Effect of chronic estradiol plus progesterone treatment on experimental arterial and venous thrombosis in mouse. PLoS One. 2017;12: 1–16. doi:10.1371/journal.pone.0177043

6. Cleuren ACA, Van Oerle R, Reitsma PH, Spronk HM, Van Vlijmen BJM. Long-term estrogen treatment of mice with a prothrombotic phenotype induces sustained increases in thrombin generation without affecting tissue fibrin deposition. J Thromb Haemost. 2012;10: 2392–2394. doi:10.1111/j.1538-7836.2012.04916.x

7. Cleuren ACA, Van Hoef B, Hoylaerts MF, Van Vlijmen BJM, Lijnen HR. Short-term ethinylestradiol treatment suppresses inferior caval vein thrombosis in obese mice. Thromb Haemost. 2009;102: 993–1000. doi:10.1160/TH09-03-0169

8. Cleuren ACA, Van Der Linden IK, De Visser YP, Wagenaar GTM, Reitsma PH, Van Vlijmen BJM. 17α-Ethinylestradiol rapidly alters transcript levels of murine coagulation genes via estrogen receptor α. J Thromb Haemost. 2010;8: 1838–1846. doi:10.1111/j.1538-7836.2010.03930.x

9. Cleuren ACA, Postmus I, Vos HL, Reitsma PH, Van Vlijmen BJM. Progestins posses poor anti-estrogenic activity on murine hepatic coagulation gene transcription despite evident anti-estrogenic activity on uterine tissue. Thromb Res. 2011;128: 200–201. doi:10.1016/j.thromres.2011.04.008

10. Kim-Schulze S, McGowan KA, Hubchak SC, Cid MC, Martin MB, Kleinman HK, et al. Expression of an estrogen receptor by human coronary artery and umbilical vein endothelial cells. Circulation. 1996;94: 1402–1407. doi:10.1161/01.CIR.94.6.1402

11. Sobrino A, Mata M, Laguna-Fernandez A, Novella S, Oviedo PJ. Estradiol Stimulates Vasodilatory and Metabolic Pathways in Cultured Human Endothelial Cells. PLoS One. 2009;4: 8242. doi:10.1371/journal.pone.0008242

---

## [Editor Report · Decision Letter 1]

28 Mar 2023

High risk oral contraceptive hormones do not directly enhance endothelial cell procoagulant activity in vitro

PONE-D-22-34362R1

Dear Drs. Bouck/Wolberg,

We’re pleased to inform you that your manuscript has been judged scientifically suitable for publication and will be formally accepted for publication once it meets all outstanding technical requirements.

Kind regards,

James P. Maloney, MD

Academic Editor

PLOS ONE

Additional Editor Comments (optional):

Thank you for submitting this excellent work.
---

## [Editor Report · Acceptance letter]

10 Apr 2023

PONE-D-22-34362R1 

High risk oral contraceptive hormones do not directly enhance endothelial cell procoagulant activity *in vitro*

Dear Dr. Wolberg:

I'm pleased to inform you that your manuscript has been deemed suitable for publication in PLOS ONE. Congratulations! Your manuscript is now with our production department. 

Kind regards, 

on behalf of

Dr. James P. Maloney 

Academic Editor

PLOS ONE